# Thinking Small: Small Molecules as Potential Synergistic Adjuncts to Checkpoint Inhibition in Melanoma

**DOI:** 10.3390/ijms22063228

**Published:** 2021-03-22

**Authors:** Alexander C. Chacon, Alexa D. Melucci, Shuyang S. Qin, Peter A. Prieto

**Affiliations:** 1Department of Surgery, University of Rochester Medical Center, Rochester, NY 14642, USA; alexander_chacon@urmc.rochester.edu (A.C.C.); alexa_melucci@urmc.rochester.edu (A.D.M.); 2Department of Microbiology & Immunology, University of Rochester School of Medicine & Dentistry, Rochester, NY 14642, USA; shuyang_qin@urmc.rochester.edu

**Keywords:** melanoma, immunotherapy, checkpoint inhibition, small molecules, targeted therapy, precision oncology

## Abstract

Metastatic melanoma remains the deadliest form of skin cancer. Immune checkpoint inhibition (ICI) immunotherapy has defined a new age in melanoma treatment, but responses remain inconsistent and some patients develop treatment resistance. The myriad of newly developed small molecular (SM) inhibitors of specific effector targets now affords a plethora of opportunities to increase therapeutic responses, even in resistant melanoma. In this review, we will discuss the multitude of SM classes currently under investigation, current and prospective clinical combinations of ICI and SM therapies, and their potential for synergism in melanoma eradication based on established mechanisms of immunotherapy resistance.

## 1. Introduction

Metastatic melanoma remains the deadliest form of skin cancer. Typified by heterogeneous lesions exhibiting variable responses to treatment, melanoma transforms from highly curable via surgical resection when localized, to hardy and recalcitrant once metastasized. Unfortunately, while the 5-year survival for localized disease less than 1.0 mm in thickness is greater than 90%, long term survival in patients with distant disease has historically been less than 10% [1,2]. In the last 15 years, the advent of targeted therapies which interact with patient-specific molecular targets—such as BRAF (B-raf proto-oncogene) inhibitors in patients with the BRAF^V600E^ point mutation—have proven efficacious in select patients with the requisite genotypes [3,4,5,6], though many patients relapse after 6–9 months of initially successful therapy [7,8]. Precision oncology—using a patient’s specific molecular profile to identify putative therapeutic targets—is emerging as the next generation of cancer care, however, many of these modalities remain preclinical or relevant for only a small selection of patients, a problem that is further elaborated on in other reviews within this issue of IJMS.

### 1.1. Current Trends in Melanoma Treatment

With the backdrop of limited responsiveness and insufficient long-term remissions induced by many cancer therapeutics, the emergence of immunotherapy has induced a paradigm shift in the approach to metastatic cancers, including melanoma. Immunotherapy, and especially immune checkpoint inhibitor (ICI) antibodies against cytotoxic T-lymphocyte antigen 4 (αCTLA-4) and programmed cell death protein-1 (αPD-1), have proven efficacious in harnessing the immune system to target tumor cells at the level of the tumor microenvironment (TME) [9]. ICIs, given as monotherapy or in combination, are now first line treatments for wild-type (non-BRAF mutant) melanoma in the adjuvant and metastatic settings, offering tremendous hope and often durable responses in some patients refractory to other forms of therapy [10,11]. Outcomes in metastatic melanoma patients are improved with ICI compared to targeted- and chemo-therapy, yet overall survival (OS) averages at less than 18 months, progression free survival (PFS) still lingers around 4–10 months, and objective response rates (ORR) are less than 40–50% [12,13]. Furthermore, pathologic complete response rates are only 13–47% for single and multi-agent regimens [14,15,16,17,18].

Beyond ICI, a variety of immunotherapeutic agents have been studied in both local and metastatic melanoma. Immunotherapy, broadly defined as any therapy which aids the immune system in combating tumor evasiveness and eliciting tumor destruction, can be broken into five main classes, with ICI and other immunomodulators being the most widely utilized. Other classes of immunotherapy include cell-based therapies (adoptive cell therapy, chimeric antigen receptor T-cell therapy), monoclonal antibodies (beyond ICI), oncolytic viruses (talimogene laherparepvec or T-VEC), and cancer vaccines. The mechanisms of these alternative forms of immunotherapy, as well as their potential synergy with small molecular drugs, are beyond the scope of this review.

ICI agents each work via inhibiting specific immune “checkpoints” which physiologically downregulate immune responses and prevent untoward immune activation. In the examples of αCTLA-4 and αPD-1, these monoclonal antibody agents block CTLA-4 (reduces T-cell priming) and PD-1 (reduces cytotoxic T-cell activation) and subsequently their downstream signaling cascades, allowing for the continued activation, expansion, and effect of tumor-specific T-cell populations [19,20]. Even with the potency of an unleashed immune system, many melanoma tumors escape antitumor immunity and develop treatment resistance, leading to the previously listed non-ideal response rates, despite multimodal therapy. Adjunct therapies to chemotherapy, targeted therapy, ICI and surgical resection look to bridge the gap from current responses to cure.

### 1.2. Small Molecule Drugs as Cancer Therapeutics

Akin to the successes of BRAF inhibitors and similar targeted drugs, an expanding type of cancer therapeutics in the form of small molecules aims to couple with existing therapies to improve response rates in metastatic melanoma and other malignancies. Small molecules (SM)—recognized as molecules <900–1000 Da in size—consist of a variety of novel and existing molecules developed or utilized for the purpose of targeting precise extra- and intra-cellular proteins, many of which are critical to biological processes such as cancer cell growth and replication. The “small” size implied by their name imparts a marked impact on the pharmacologic properties of SMs, especially regarding drug administration and cellular penetration [21]. SMs can translocate through plasma membranes, affording access to targets which monoclonal antibodies and similar “large” therapeutics cannot. Many SMs are also amenable to oral administration, while antibody-based treatments typically require intravenous administration. The comparatively shorter half-lives and linear clearance of SMs potentially reduce the risk of systemic toxicity compared to other drug classes. Furthermore, many SMs are far less expensive and require fewer resources to develop, however, they are limited by less specific binding, and thus, often exhibit the potential for more widespread off-target effects [22,23].

As regimens continue to be refined regarding scheduling, dose, and combinations of chemo- and immunotherapy agents for metastatic melanoma, it is SMs which may afford the most potential for combinatorial therapies given their ease of production, administration, and broad mechanistic reach. Furthermore, the heightened infiltrative and destructive T-cell function proffered by ICI may prove to be an ideal partner for drugs which can expose tumor neoantigens and recruit adaptive effectors. In this review, we will discuss the potential synergy of small molecular drugs and immune checkpoint inhibition in the treatment of metastatic melanoma. We will highlight the successes and pitfalls of clinical studies combining SMs and ICIs, discuss promising combinations utilized in preclinical settings, and propose potential mechanisms of synergy between these disparate therapeutic modalities in combating tumor immune evasion and treatment resistance.

## 2. Small Molecule Drugs and Potential Synergy with Immune Checkpoint Inhibition

### 2.1. Mechanisms of Immunotherapy Resistance

In order to discuss the potential synergy of SMs and ICI in treating melanoma, it is paramount to understand the mechanisms of immune escape which result in treatment failure. Resistance to ICI can be primary (completely refractory) or acquired/secondary (initial response and then relapse); working definitions of clinical resistance are still being defined [24]. While the molecular mechanisms behind these types of treatment resistance are cancer-specific, treatment type-specific, and are still being elucidated, a variety of contributors to resistance patterns have been postulated and are reviewed in detail in a previous issue of *The International Journal of Molecular Sciences* [25]. Briefly, known factors include the tumor mutational burden, immune and metabolic factors from the tumor microenvironment, genomic drivers within tumor cells, and host factors, among other contributors (Figure 1).

In melanoma, the tumor mutational burden and clonal neoantigen burden have been associated with the robustness of ICI response [26,27,28,29], while a hyper-mutational phenotype has been correlated with durable responses to αPD-1 therapy in non-small cell lung cancer [30]. Thus, in theory, given the high mutational burden of melanomas, strategies to increase exposed melanoma neoantigens may prove especially effective in increasing responses to therapy [31,32]. This strategy, however, relies on the assumption that the number of exposed neoantigens correlates directly with the robustness of immune infiltration (and T-cell infiltration) into the TME, an incompletely understood phenomenon with variability across cancer types.

The melanoma TME, consisting of a complex system of tumor cells, stromal cells, immune cells, metabolic infiltrates, and all other intra-tumoral components and interactions, also has a profound impact on ICI responsiveness. The role of stromal cells in promoting tumor immune escape and resistance to therapy is reviewed in detail by Mazurkiewicz et al. in this issue of IJMS (Mazurkiewicz IJMS 2021). The effect of immune populations within the TME in clearance of melanocytic cells is intricate, multifaceted, and appears dependent on the function of cytotoxic T-cells [33]. Specifically, the ability of metabolic and immune factors to suppress CD8 T-cell infiltration and function appears vital in promoting tumor immune escape [34], and higher numbers of pre-treatment CD8 expressing T-cells at the tumor margin is predictive of the response to αPD-1 in human melanoma [35]. Furthermore, tumor cell-specific activation of the WNT β-catenin pathway has been correlated with absence of T-cell infiltrate in metastatic melanoma, purported to contribute to ICI resistance [36]. The presence of CD4^+^ T-regulatory cells (Treg), myeloid derived suppressor cells (MDSCs), tumor associated macrophages (TAMs), and cancer associated fibroblasts (CAFs) has been associated with poor prognosis in a variety of cancers due to their relationship with impaired functional cytotoxic T-cell infiltration [25,37,38]. Creation of a “hot” TME—characterized by an absence of Tregs and/or MDSCs and an abundance of both tumor cells expressing checkpoint molecules and cytotoxic T-cells—remains an elusive target, even in theoretically more immunogenic cancers such as melanoma. Checkpoint inhibitors themselves may elicit a reduction in MDSCs, however, maintenance of an environment rich in tumor-ablative immune cells or de novo creation of this environment from a “cold” TME is essential in ensuring adequate responsiveness to ICI therapy [39,40].

As discussed above, the presence of tumor neoantigens and subsequent immune recognition of tumor cells is essential for tumor clearance. Loss of these neoantigens through tumor-intrinsic genomic factors may be a mechanism of immune and treatment escape in melanoma. For example, mutations in the JAK/STAT pathway abrogate responsiveness to IFNγ signaling and reduce neoantigen presentation, leading to a lack of responsiveness to ICI [40,41]. TGFβ-driven transcriptional downregulation of MHC class I has recently been found to be a hallmark of resistance to αPD-1 ICI in melanoma [42]. Mutations in canonical signaling pathways (MAPK-ERK and PI3K) can also impact immune recruitment, while mutations in angiogenic factors may similarly alter components of the TME, with subsequent impacts on responsiveness to ICI [43].

Lastly, a variety of host factors including: immune genetics, such as HLA genotypes [44]; metabolic factors; other drugs or therapies, including antibiotics [45,46,47]; in addition to the patient’s gut microbiome [48,49,50], have a marked impact on the responsiveness to checkpoint therapy. While the myriad of techniques to manipulate these factors is beyond the scope of this review, it is vital to note that a wide array of potential avenues exist to augment baseline immune function and subsequent responsiveness to immune checkpoint inhibition. The potential for SMs—many of which have a direct effect on the modalities of immune escape listed above—to synergize with ICI and potentiate its effects are multifold.

### 2.2. Classification of Small Molecule Drugs

Small molecules can be classified according to a variety of schemas, including by molecular mechanism of action (e.g., growth factor kinase inhibitor, telomerase inhibitor, etc.), by overarching physiologic effect (e.g., anti-angiogenic, immunomodulatory, etc.), by chemical composition/structure, or otherwise. For the purposes of this review, we will classify SMs by their broad mechanistic effect, in order to assess the potential synergy of individual SMs with ICI agents. A detailed list of recently completed, ongoing, and planned clinical trials combining an SM agent with ICIs in melanoma is presented in Table 1.

### 2.3. BRAF/MEK Inhibitors

To date, the most widely studied combinations of SMs and ICIs include the family of “targeted therapy” drugs which inhibit rapidly accelerated fibrosarcoma (RAF) and mitogen activated protein kinase kinases (MEK). The canonical mitogen-activated protein kinase (MAPK) signal transduction pathway of receptor tyrosine kinase—RAS—RAF—MEK—ERK is conducted within the cytoplasm and involved in cell survival, growth, migration, and apoptotic resistance. Approximately 40–60% of cutaneous melanoma patients harbor mutations in the BRAF gene (most commonly the BRAF^V600E^ point mutation) [77], and small molecular BRAF inhibitors (BRAFi) targeting this serine/threonine protein kinase (vemurafenib, dabrafenib, and encorafenib) are now used as first line treatments in metastatic BRAF-mutant melanoma [2]. Combination with MEK inhibitors potentiates the effect of RAF inhibitors, and have proven to be efficacious in the clinical treatment of late-stage melanoma [8]. Unfortunately, while many patients have initially potent responses to these drugs, a majority will develop resistance to combined BRAF/MEK inhibition [7]. Thus, combination with other therapies—especially immune-based therapies such as ICI—has been sought after to potentially synergize with the cancer cell death (and subsequent antigen presentation and T-cell infiltration) induced by RAF and MEK inhibitors [78,79].

Initial clinical studies addressed the efficacy of BRAFi + αCTLA-4, with subsequent studies assessing the efficacy with αPD-1. The initial phase 1 clinical trial, combining single agent BRAFi (vemurafenib) with single agent αCTLA-4 (ipilimumab), was unfortunately terminated early due to hepatotoxicity with concurrent dosing [51]. A subsequent trial of an alternative sequencing strategy of these agents showed an improved safety profile, however, the median OS was only 18.5 months and progression free survival (PFS) was only 4.5 months [52]. Additionally, the combination of the second generation BRAFi/MEKi dabrafenib and trametinib with ipilimumab was noted to have significant GI toxicity [57]. In the KEYNOTE-022 trial, a randomized phase 2 trial studying dabrafenib, trametinib and αPD-1 (pembrolizumab), the combination was slightly better tolerated, with a 73% ORR and median PFS of 16.9 months [58,59,60]. This combination was more tolerable when dabrafenib and trametinib were given intermittently in early results from the ImPemBra trial [61]. The combination of dabrafenib, trametinib, and nivolumab (αPD-1) resulted in an 89% ORR in early results of a single arm phase 2 study [62]. Additionally, the combination of dabrafenib, trametinib, and spartalizumab (αPD-1) is also promising in early trial results, with the phase 3 COMBI-I trial eliciting an ORR of 78% [29]. Finally, the combination of the newer BRAFi/MEKi agents, encorafenib and binimetinib, utilized in a sequential design with ipilimumab plus nivolumab (randomized to one treatment modality until progression of disease, then switch), is still ongoing in the SECOMBIT trial, with early ORR of 82.6% and median PFS of 15.8 months when targeted therapy was given first, and poorer responses when ICI was given first [64]. While it appears that BRAFi/MEKi combination with αPD-1 ICI offers significant benefits over ICI monotherapy, increased toxicity over BRAFi and ICI monotherapies are noted, and thus, completion of these trials will allow for more detailed analysis of the safety, efficacy, and mechanistic synergy of these regimens.

Promising results have been seen in a phase 1b study combining atezolizumab (monoclonal antibody against PD-ligand 1, or αPD-L1) with vemurafenib and cobimetinib (MEKi) after a 28-day run-in period with vemurafenib plus cobimetinib, as the ORR was 71.8% and there was a median response duration of 17.4 months (phase 3 trial ongoing) [53]. A similar randomized trial assessing triple therapy with atezolizumab, vemurafenib, and cobimetinib showed increased PFS to 15.1 months vs. 10.6 months with a placebo instead of atezolizumab, and was well tolerated [54]. In the melanoma arm of a phase 1b trial of patients with advanced solid tumors, the combination of cobimetinib and atezolizumab resulted in an ORR of 41% in a mixed BRAF mutant/wild-type cohort [55]. The combination of αPD-L1 and SMs is also under investigation in a number of other clinical trials (Table 1). Similar results have been found in a phase 1, open-label, dose-escalation and -expansion study using durvalumab (αPD-L1) with dabrafenib and trametinib, with an ORR of 69.2% for BRAF mutant patients, but less than 32% for wild-type patients [65].

When weighing BRAFi/MEKi and ICI combination strategies, it is important to consider the relatively poorer ICI responsiveness in patients who have previously failed BRAFi/MEKi targeted therapy. In a study of pembrolizumab, nivolumab, or nivolumab plus ipilimumab in patients who previously failed targeted therapy, the ORR was only 18% with αPD-1 treatment and 15% in αPD1/αCTLA4 treatment [63]. However, due to the improvements over previous clinical outcomes as well as the multitude of BRAFi, MEKi, and ICI combinations, there remains significant cause for excitement about the potential synergy of these drugs for clinical benefit. Multiple combinations of SM BRAFi/MEKi and ICI immunotherapy are being studied in ongoing or planned clinical trials, which can be found in Table 1.

### 2.4. Oncogenic Driver Inhibitors and Kinase Inhibitors

Apart from the MAPK pathway, a variety of oncogenic drivers implicated in signal transduction and proto-oncogene function are attractive targets for putative synergy with ICI immunotherapy. As melanoma has the highest mutational frequency of any malignancy [31], multimodal targeting may offer a beneficial strategy in combating the multitude of treatment escape mechanisms in metastatic melanoma. The current generation of oncogenic driver inhibitor drugs act via interrupting constitutively active signaling cascades, which leads to the aberrant growth and replication of tumor cells. Small molecules targeting tyrosine kinases (TKI, typically inhibitory) are the most extensively studied of these drugs, however, many have broad off-target effects. For example, imatinib, a receptor TKI utilized to target BCR-ABL in gastrointestinal stromal tumor treatment, also has off-target effects on at least C-KIT, PDGFR, and DDR1. Furthermore, inhibition of specific signaling cascades in malignant cells often allows for collateral escape and continued function akin to the resistance mechanisms seen in BRAF inhibition [80], and thus, pleiotropic drugs may afford the potential for increased durability. Pro-neoplastic oncogenic driver targets involved in cell proliferation include phosphatidlyinositol-3-kinase (PI3K), bruton tyrosine kinase (BTK), MET proto-oncogene, and cyclin dependent kinases (CDKs), among many others [81,82,83,84,85]. While the plethora of oncogenic driver inhibitors act via varied mechanisms, ultimately, their targets include pathways which disrupt growth and replication (see above) or aberrant differentiation (e.g., the hedgehog pathway, cereblon, ATR3), allowing for increased drug efficacy in rapidly dividing tumor cells. As with RAF inhibitors, putative mechanistic synergy exists between inhibition of oncogenic signaling pathways and immunotherapy in the cessation of tumor cell growth. Subsequent tumor cell death may lead to increased tumor neoantigen presentation, increased innate immune cell recruitment to the TME, alteration in proinflammatory cytokines, and subsequent recruitment of cytotoxic T-cells.

Similar to early clinical trials combining BRAFi and ICI, early efforts combined SMs targeting oncogenic drivers and TKIs with αCTLA-4. A phase 1 dose-escalation study of the TKI imatinib and ipilimumab in gastrointestinal stromal tumors and melanoma showed low activity with no clear signal for synergy [67]. A subsequent phase 2 study of imatinib and pembrolizumab was withdrawn due to poor study accrual (NCT02812693). Clinically assessed combinations in this class are fairly limited. Ongoing studies assessing a variety of other oncogenic driver and TKI targets in combination with ICI, and especially with αPD-1 therapies, can be found in Table 1.

### 2.5. Anti-Angiogenic Molecules

When melanoma reaches a size threshold for nutrient diffusion, a tumor-specific neoplastic vasculature of aberrant and incomplete vessels propagates in a process called neoangiogenesis, allowing for continued blood and nutrient supply to highly metabolic tumor cells. Inhibition of angiogenesis with monoclonal antibodies such as bevacizumab in combination with ICI has been studied with only moderate clinical responses [86,87], and continues to be evaluated in multiple clinical trials (NCT03175432). However, angiogenesis-inhibiting SMs may proffer different results due to their ability for cellular penetration. Many SMs targeting angiogenesis act on tyrosine kinase receptors, such as the vascular endothelial growth factor receptor (VEGFR) or the platelet derived growth factor receptor (PDGFR) families, while other novel agents act on angiopoietin 1 and 2, or other endothelial factors. These factors are upregulated in the endothelium, especially in newly proliferating melanoma tumors, resulting in both neovascularization from existing vessels and recruitment of bone marrow progenitors which reach hypoxic regions and induce de novo vessel formation [88,89,90]. Blocking neoangiogenesis may limit further tumor growth or even result in impaired nutrient delivery and tumor cell apoptosis/necrosis, resulting in tumor antigen exposure and subsequent immune recruitment. Conversely, impaired blood flow may inhibit the infiltration or function of the cytotoxic T-cells promoted by ICI, an effect which is actively being studied (Table 1) and remains to be defined.

Initial studies combining anti-angiogenic SMs with ICI have investigated drugs which inhibit the VEGFR family of receptors, including axitinib, lenvatinib (multi-kinase inhibitor), and apatinib. In a phase 1b study evaluating axitinib with toripalimab (αPD-1) in mucosal melanoma, which tends to be less responsive to αPD-1 monotherapy than cutaneous melanoma, an ORR of 48.3% was seen in a cohort of 33 Chinese patients [69]. In early results from the LEAP (LEnvatinib And Pembrolizumab) 004 trial, a phase 2 study of patients who previously progressed on αPD-1/L1 therapy, the combination of lenvatinib and pembrolizumab resulted in ORR of 31%, with a safety profile similar to prior ICI monotherapy [70]. While these small study results are preliminary and conducted in unique populations, their results demonstrate significant potential for anti-angiogenic SMs as an adjunct to ICI immunotherapy, especially in the rescue setting. Further ongoing clinical trials, the majority of which combine VEGFR inhibitors with αPD-1 therapies, can be found in Table 1.

### 2.6. Epigenetic Modifiers

Epigenetic modifications (alterations to gene expression resulting from DNA methylation, histone modification, regulatory non-coding RNA, or transcription factors) are highly prevalent in cancers, including in melanoma [91]. DNA methylation, histone acetylation, and histone methylation represent the most accessible targets for SMs in cancer treatment, while non-epigenetic agents targeting DNA alkylation, nucleoside incorporation, and poly ADP ribose polymerase inhibitors may provide direct genetic effects on tumors that result in increased tumor cell death. Epigenetic alterations within tumors may create mechanisms for immune escape (and treatment resistance) [81,92,93]. In fact, preclinical models show that pharmacologic epigenetic modulation of tumor cells may overcome αPD-1 resistance through increased T-effector cell infiltration in B16 melanoma [94]. Targeting of epigenetically silenced pathways within tumor cells may expose previously unexpressed neoantigens on the tumor surface, affording immune recognition and clearance [95]. Furthermore, it is feasible that epigenetic modification in immune cells may promote more robust activation of innate and adaptive immune effectors, as highlighted elsewhere [96,97]. Augmentation of immunity through epigenetic alterations could potentiate ICI-initiated T-cell function against cancer antigens (with the caveat that this mechanism may promote over-activation and cross-antigen reactivity).

Clinical studies utilizing epigenetic modifiers are just beginning to be completed. In the PEMDAC (Pembrolizumab with Entinostat to Treat Metastatic Melanoma of the Eye) trial, a phase 2 open label study assessing the histone deacetylase inhibitor entinostat with pembrolizumab in patients with metastatic uveal melanoma (59% of which had received previous treatment), an ORR of 10% was observed, with a median OS of 11.5 months [71]. In the NIBIT-M4 (Italian Network for Tumor BIoTherapy – Metastatic Melanoma) trial, a phase 1b study of the DNA hypomethylating agent guadecitabine with ipilimumab, the ORR was 26%, with tumor immune contexture showing an increase in CD8^+^, PD-1^+^ T-cells in post-treatment tumors [72]. Further studies which address the utility of epigenetic modifiers, and especially histone deacetylase inhibitors, with ICI immunotherapy can be seen in Table 1.

### 2.7. Dual Immunomodulation

Perhaps the most promising methods to enhance ICI immunotherapy efficacy are combinations with agents which further augment or alter immune function. These drugs, known as immunomodulators, represent a diverse array of therapies which may alter immune populations, inhibit or potentiate immune effector functions, or regulate immunity using other novel strategies. As a vast array of immune populations and subpopulations exist, an equally large number of potential targets exist for SM binding. The most widely assessed targets include those which regulate myeloid infiltration or proliferation, the JAK/STAT pathway, and the indoleamine 2,3-dioxygenase 1 enzyme (IDO). Specific emphasis continues to be placed on the role of MDSC-induced suppression of T-cell recruitment and effector function, as discussed above. While ICI are not without immune related toxicities, the possibilities of either further augmenting T-cell function through additional T-cell activation/propagation or targeted inhibition of immunosuppression are enticing putative mechanisms for improving non-ideal response rates of current standard of care regimens.

Initial excitement over the potential for synergism between secondary immunomodulators and ICI took flight with the use of adjuvant IDO inhibitors (implicated in inhibiting T-cell proliferation). The combination of indoximod and investigator’s choice of ICI (pembrolizumab, nivolumab, or ipilimumab) resulted in a 55.7% ORR for indoximod plus pembrolizumab [74]. Unfortunately, momentum was allayed after the results of the phase 3, double-blind randomized KEYNOTE-252 trial combining epacadostat plus pembrolizumab showed a lack of synergy and no improvement over single agent pembrolizumab [73], subsequently resulting in termination of other trials (NCT03361228). Interestingly, the PI3K-γ inhibitor eganelisib may reprogram MDSCs towards immune activation and/or provide the metabolic stress needed to induce tumor cell death in concert with ICI. In the 40-patient melanoma expansion arm of an ongoing phase 1/1b study of eganelisib and nivolumab in patients who had previously failed αPD-L1 therapy, a small subset of patients who had received ≤ two lines of prior systemic therapy experienced an ORR of 15.8% [75].

Within the TME, significant potential exists for synergy between SMs which alter other immune populations and ICI in melanoma. Inhibition of the colony stimulating factor-1 receptor, chemokine receptor CXCR1 and 2, the JAK/STAT pathway, and other modalities are currently undergoing clinical trial and are detailed in Table 1. SMs which alter TME infiltration and effector function of dendritic cells, natural killer cells, components of the mono-nuclear phagocyte system, MDSCs, and other immune cell compartments, may promote an acute inflammatory or immunoreactive environment favorable for tumor cell clearance [98]. Alternatively, modification of non-immune TME cellular compartments may, in turn, have indirect effects on antigen presenting cells and downstream anti-tumor effectors such as natural killer cells or CD8 T-cells. As noted above, SMs of many classes including TKIs and epigenetic modifiers all have the potential for broad effects on tumor, immune, and other TME components. More directly, stimulator of interferon genes (STING) agonists in concert with ICI are an exciting putative combination due to the potential for *de novo* immune activation and increased CD8 T-cell priming. While dual immunomodulation represents both an enticing and sensible strategy for synergy in melanoma, much of this work is preclinical and in early phase clinical trials, and is reviewed elsewhere [98,99].

### 2.8. Novel Checkpoint Therapies on the Horizon

Apart from the currently established therapies of αCTLA-4, αPD-1, and αPD-L1, new checkpoint inhibitors have been and continue to be developed for the purposes of treating metastatic cancers including melanoma. These include antibodies against the checkpoint molecules LAG-3, TIM-3, and TIGIT, among others [100]. Furthermore, a new type of antibody with bi-specificity for dual checkpoint inhibition may prove promising. Clinical trials utilizing bispecific antibodies against CTLA-4 and PD-1 (SI-B003, NCT04606472), LAG3 and PD-1 (Tebotelimab, NCT04653038), and TIM-3 and PD-1 (RO7121661, NCT03708328) are in the early stages of recruitment. Furthermore, SM checkpoint inhibitors are currently being developed and tested [101,102], exemplified by an ongoing phase 2 study of an oral PD-L1 inhibitor (INCB086550, NCT04629339). The safety and clinical benefit of these drugs remains to be seen. However, novel modes of checkpoint inhibition may either potentiate or prove superior to existing therapies, with the benefit of adjunctive therapies, including SM drugs, to be tested thereafter.

### 2.9. Other Potential Synergistic Therapies, Including Nutritive Therapies

A variety of other SM therapies may provide synergistic benefit when combined with ICI, on the basis of their known relationships with melanoma disease progression. These alternative SMs include fructose or other small carbohydrates, lipid-based therapies including short chain fatty acid derivatives, and other drugs with alternative targets such as metformin (and analogues) and non-steroidal anti-inflammatory drugs (NSAIDs). With broad effects including platelet aggregation, prostaglandin mediator production, and mitochondrial uncoupling, aspirin and other NSAIDs, and particularly cyclooxygenase (COX) inhibitors, are garnering much attention due to their mechanistic sensibility and apparent clinical benefit in improving ICI efficacy. The use of COX inhibitors concurrently with ICI treatment has been associated with longer time to disease progression and improved the ORR at 6 months in a cohort of melanoma and non-small cell lung cancer patients [45]. In melanoma models, the COX2 pathway contributes to tumor immune evasion through the effects of prostaglandin E2, and inhibition of this pathway has led to synergy with αPD-1 in a preclinical setting [34,103,104].

As mentioned above in the section on mechanisms of immunotherapy resistance, the gut (and likely tumor) microbiome plays a significant role in responsiveness to therapy. Specifically, the presence of a particular gut microbiome may portend enhanced systemic and antitumor immunity in melanoma patients responsive to αPD-1 therapy [49,50]. While the specific bacterial (or viral) genera which elicit differential treatment responsiveness is still under investigation, both gut and tumor microbial abundance, diversity, and functional metabolism appear central in contributing to immune stimulation and activation [105,106]. Thus, SM antibiotics, prebiotics, or nutritive therapies which directly and indirectly modulate the microbiome are likely to significantly impact the efficacy of ICI immunotherapy in melanoma. Interestingly, cumulative antibiotic use during the period immediately before, during, and after treatment with ICI has been associated with worsened OS and PFS in patients with advanced cancers (including melanoma) [47], and it may be that maintenance of a diverse and rich microbiome is necessary for optimal anti-tumor immunity. Nutritive therapies, in particular, have the potential for multimodal benefit, simultaneously affecting all aspects of the TME and with putative impacts on host immunity, microbial physiology, tumor growth, and immune evasion [107]. In part due to the complexity of nutritional physiology, marked heterogeneity exists in study design of many nutritional interventions, complicating the determination of success of specific nutritive therapies. While enticing mechanistically, the efficacy of many alternative therapies in concert with ICI is theoretical or preclinical in nature and remains to be investigated clinically.

## 3. Discussion

Marked progress has been made in melanoma treatment over the last two decades. The advent of precision oncology, highlighted by targeted therapies with patient-specific or tumor-specific effects, has ushered in a new era of cancer care. Coupled with the emergence of checkpoint inhibitor immunotherapy, cure rates are significantly improving even in metastatic disease, yet more than half of patients will ultimately not respond to therapy. Thus, the discovery and clinical evaluation of new treatment regimens for those with widespread, recalcitrant, or resistant disease remains of utmost importance. SMs, spanning a wide spectrum of molecular targets, offer a plethora of opportunities for mechanistic synergy with ICI immunotherapy, many of which are already being evaluated in melanoma patients.

In this review, we have highlighted the potential synergy between well-studied SM classes and ICI for the treatment of metastatic melanoma. BRAF/MEK inhibition, oncogenic driver and tyrosine kinase inhibition, inhibition of angiogenesis, epigenetic modulation, and immunomodulation represent only a portion of the possible SM/ICI combinations being evaluated in cancer treatment, yet illustrate the mechanistic rationale for novel drug combinations utilized to combat metastatic melanoma (Figure 2). To improve ICI efficacy, SMs may act on tumor cells, immune effectors, or other TME components to synergize with T-cell activation induced by ICI. Whether by directly eliciting tumor cell death (or increasing neoantigen presentation at the tumor cell surface) or by fortifying immune, metabolic and structural TME components which afford immunogenic tumor clearance, SMs hold the potential for augmenting the biologic efficacy of ICI in the melanoma TME and improving clinical responses in melanoma treatment.

Due to the immense number of available targets, alternative SMs which act in the TME and are not reviewed herein, may indeed prove efficacious in melanoma treatment [108]. SMs targeting matrix metalloproteinases, heat shock proteins, proteosome components, and many other targets may provide tumor-killing and immune-activating benefit in concert with ICI.

Furthermore, novel clinical and laboratory techniques are affording a new generation of diagnostic and prognostic strategies in clinical melanoma treatment. The increased availability of DNA sequencing techniques has altered the therapeutic landscape in clinical oncology, now allowing identification of patient-specific (somatic) or oncogenic driver genetic mutations (akin to BRAF^V600E^ mutant) which may act as biomarkers in the development of future targeted therapies. Apart from those discussed in this review, these include C-KIT and NRAS among other targets, and a new generation of targeted oncogenic driver inhibitors may promote more precise treatment algorithms in melanoma [109]. Furthermore, neoantigens exposed after treatment with targeted therapy or immunotherapy may alter the clinical approach to treatment. Tumor neoantigen production is indicative of durable response to αPD-1 therapy in other cancers including non-small cell lung cancer [30,110], and there is an increasing appreciation for their possible role as a predictive biomarker in melanoma [29,111]. Unearthing the utility of novel biomarkers in dictating treatment algorithms and improving responses in metastatic melanoma is an exciting new endeavor. It is likely that newly discovered biomarkers of melanoma progression and of response to ICI immunotherapy will reveal new potential combinatorial targets to improve treatment efficacy even in highly resistant disease.

Importantly, combination therapies may also create the potential for additional toxicities beyond those associated with ICI or a specific SM alone, especially hepatotoxicity and GI toxicity [51,57]. This effect may be even more pronounced for potent SMs than for other therapeutic modalities. Failure to reach the recommended phase 2 doses in clinical trials combining ICI and second agents (chemotherapy, monoclonal antibodies, SMs, vaccines, viruses, dendritic therapies) has been associated with the choice of second agent, with higher failure rate (53%) for small molecular drugs than other agent types [112]. While the generally increased safety profile of αPD-1 agents over αCTLA-4 agents has been encouraging, the improved but imperfect benefit noted with standard-of-care dual ICI favors a continued search for adjunctive therapies for patients with advanced disease, even at the risk of increased toxicities. Alternative dosing and treatment sequences (such as neoadjuvant vs. adjuvant ICI in patients with resectable locally advanced disease) may also provide insight into new treatment algorithms with altered safety and efficacy profiles among specific patient groups [113].

## 4. Conclusions

In the current era of melanoma treatment, novel techniques in diagnostics, drug development, and therapeutic delivery have paved the way for a precision oncology approach to cancer care. While immunotherapy agents including immune checkpoint inhibitors have revolutionized the treatment of metastatic melanoma, response rates still remain inadequate for the majority of patients. Innovative drug therapy combinations, including the addition of small molecule drugs with targeted mechanisms of action, may offer the potential for modes of synergy with current and future checkpoint inhibitor regimens. Until improved treatment regimens are discovered, creative combinatorial strategies should be trialed in an attempt to approach cure for metastatic melanoma.

## Figures and Tables

**Figure 1 ijms-22-03228-f001:**
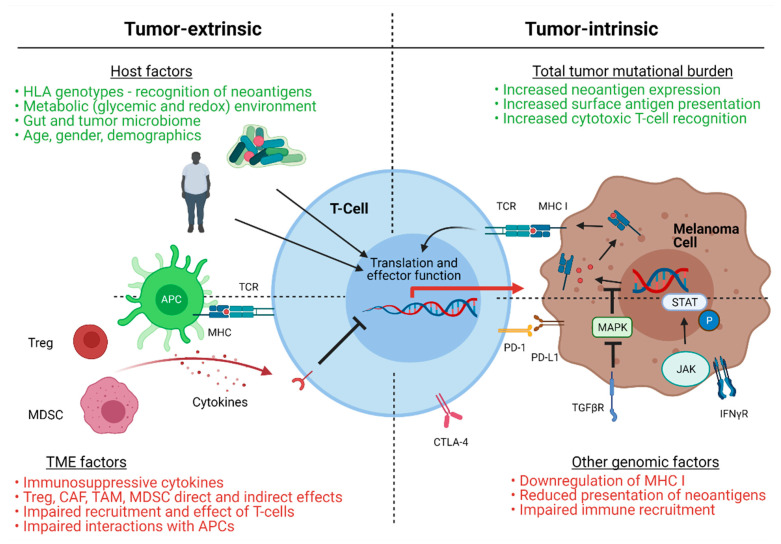
Proposed mechanisms of resistance to immune checkpoint inhibition in melanoma. In green are factors associated with potential immune checkpoint inhibitor responsiveness, while red highlights factors associated with potential resistance to therapy. Tumor extrinsic mechanisms include host factors and tumor microenvironment (TME) factors. Host factors include: immune recognition via specific human leukocyte antigen (HLA) genotypes, metabolic factors, such as obesity and diabetes; the gut/tumor microbiome; demographics, such as age or gender. TME factors include: an immunosuppressive cytokine milieu; the presence of regulatory T-cells (Treg); cancer associated fibroblasts (CAF); tumor associated macrophages (TAM)); myeloid derived suppressor cells (MDSC) and other cell types. Each elicits specific effects on T-cell recruitment and activity, and may impair interactions between antigen presenting cells (APCs) and T-cells. Tumor-intrinsic mechanisms include overall tumor mutational burden and other tumor genomic factors. High tumor mutational burden increases neoantigen expression and surface antigen presentation for recognition by activated cytotoxic T-lymphocytes, augmented with the use of immune checkpoint inhibition. Tumor cell mutation of specific oncogenic drivers or signaling pathways results in altered responses to interferon gamma and reduced presentation of neoantigens on major histocompatibility complex (MHC) I, ultimately altering immune effector recruitment and activation. TCR (T-cell receptor); PD-1 (programmed cell death protein-1); PD-L1 (programmed death-ligand 1); CTLA-4 (cytotoxic T-lymphocyte antigen 4); MAPK (mitogen activated protein kinase); STAT (signal transducer and activator of transcription); JAK (Janus kinase); TGFβR (transforming growth factor beta receptor); INFγR (interferon gamma receptor); P (phosphorylated).

**Figure 2 ijms-22-03228-f002:**
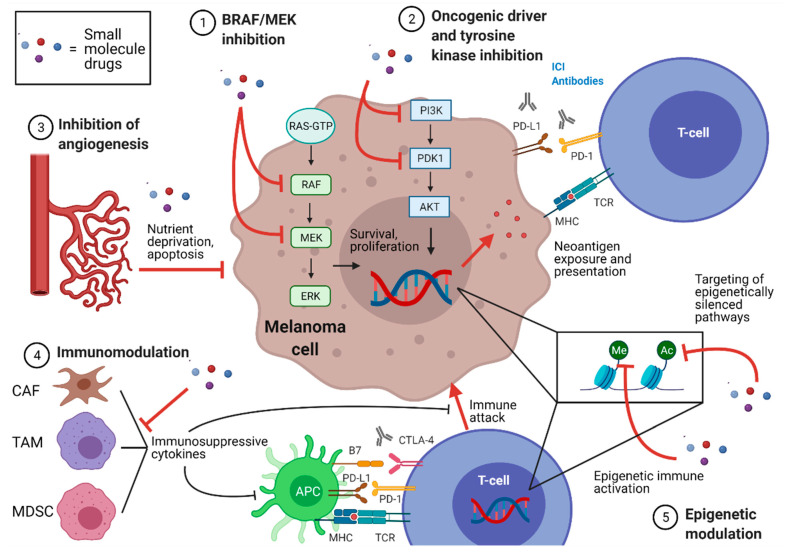
Potential mechanisms of synergy between small molecules and immune checkpoint inhibition in melanoma. Colored dots represent small molecule drugs. Intracellular ablation of Braf proto-oncogene/mitogen-activated protein kinase kinase (BRAF/MEK) in the mitogen activated protein kinase (MAPK) pathway (1), of oncogenic drivers such as those involved in the phosphoinositide-3-kinase/protein kinase B (PI3K/AKT) pathway or of other tyrosine kinases (2) induce apoptosis, resulting in increased presentation of neoantigens at the cell surface to primed and activated T-cells in the tumor microenvironment (TME). Extrinsic to the tumor cell, anti-angiogenic drugs (3) may impair nutrient delivery to tumor cells, resulting in tumor cell death and immune clearance. In the immune axis, immunomodulator drugs (4) may impair immunosuppressive cell types such as cancer associated fibroblasts (CAF), tumor associated macrophages (TAM), and myeloid derived suppressor cells (MDSC), including the cytokines they secrete which hinder cytotoxic T-cell function. Epigenetic modulator drugs may have bi-fold effects (5), exposing epigenetically silenced pathways (methylated, Me, or acetylated, Ac) within tumor cells, or alternatively activating pathways within immune effector cells including but not limited to immune checkpoints. APC (antigen presenting cell); MHC(major histocompatibility complex); RAS-GTP (Ras proto-oncogene guanosine triphosphate); ERK (extracellular signal-regulated kinases); PDK1 (phosphoinositide dependent kinase 1); TCR (T-cell receptor); PD-1 (programmed cell death protein-1); PD-L1 (programmed death-ligand 1); CTLA-4 (cytotoxic T-lymphocyte antigen 4).

**Table 1 ijms-22-03228-t001:** Completed and ongoing clinical trials combining small molecules and checkpoint inhibitors in malignant melanoma.

Small Molecule Agent	Molecular Targeting Mechanism	Immunotherapy Agent(s)	Cancer Type	Reference	Status	Phase	Outcome/Anticipated Date
Vemurafenib	BRAF inhibitor	Ipilimumab	Melanoma	NCT01400451 [51]	Terminated	1	Hepatotoxicity with concurrent dosing
Vemurafenib	BRAF inhibitor	Ipilimumab	Melanoma	NCT01673854 [52]	Completed	2	Median PFS 4.5 months, improved safety vs. concurrent administration
Vemurafenib + Cobimetinib	BRAF-MEK inhibitors	Nivolumab + Ipilimumab	Melanoma	NCT02968303	Recruiting	2	10/2023
Vemurafenib + Cobimetinib	BRAF-MEK inhibitors	Nivolumab + Ipilimumab	Melanoma	NCT02968303	Recruiting	2	6/2020
Vemurafenib + Cobimetinib	BRAF-MEK inhibitors	Prior first-line immunotherapy	Melanoma	NCT03224208	Recruiting	2	12/2022
Vemurafenib + Cobimetinib	BRAF-MEK inhibitors	Pembrolizumab	Melanoma	NCT02818023	Active, not recruiting	1	5/2024
Vemurafenib +/− Cobimetinib	BRAF-MEK inhibitors	Atezolizumab	Melanoma	NCT01656642 [53]	Completed	1b	ORR 71.8%, substantial but manageable toxicity
Vemurafenib + Cobimetinib	BRAF-MEK inhibitors	Atezolizumab	Melanoma	NCT02902029	Active, not recruiting	2	6/2022
Vemurabinib + Cobimetinib	BRAF-MEK inhibitors	Atezolizumab	Melanoma	NCT02908672—TRILOGY [54]	Active, not recruiting	3	Median PFS 15.1 vs. 10.6 months in triple therapy vs. without atezolizumab
Vemurafenib + Cobimetinib	BRAF-MEK inhibitors	Atezolizumab	Melanoma	NCT04722575	Recruiting	2	6/2027
Vemurafenib + Cobimetinib	BRAF-MEK inhibitors	Atezolizumab	Melanoma	NCT02303951	Terminated	2	Low recruitment
Vemurafenib + Cobimetinib	BRAF-MEK inhibitors	Atezolizumab	Melanoma	NCT03554083	Recruiting	2	6/2023
Cobimetinib	MEK inhibitor	Atezolizumab	Advanced solid tumors	NCT01988896 [55]	Completed	1	ORR 41% in mixed BRAF mutant/WT population, median PFS 12 months
Cobimetinib	MEK inhibitor	Atezolizumab vs. Pembrolizumab	Melanoma	NCT03273153—IMspire 170 [56]	Active, not recruiting	3	3/2025 Early results—median PFS 5.5 months for cobimetinib + atezolizumab, 5.3 months for Pem
Dabrafenib	BRAF inhibitor	Ipilimumab	Melanoma	NCT02200562	Terminated	1	Support withdrawn
Dabrafenib +/− Trametinib +/− Nivolumab	BRAF-MEK inhibitors +/− αPD-1	Ipilimumab	Melanoma	NCT01940809	Active, not recruiting	1	7/2020
Dabrafenib +/− Trametinib	BRAF-MEK inhibitors	Ipilimumab	Melanoma	NCT01767454 [57]	Completed	1	Initially safe, 2/7 patients on triple therapy with colitis/perforation
Dabrafenib +/− Trametinib	BRAF-MEK inhibitors	Ipilimumab and Nivolumab	Melanoma	NCT02224781—DREAMSEQ	Recruiting	3	10/2022
Dabrafenib + Trametinib	BRAF-MEK inhibitors	Ipilimumab + Nivolumab	Melanoma	NCT02224781	Recruiting	3	10/2022
Dabrafenib + Trametinib	BRAF-MEK inhibitors	Pembrolizumab	Melanoma	NCT02130466—KEYNOTE-022 [58,59,60]	Active, not recruiting	2	Median PFS 16 months on triple therapy vs. 10.3 months without, 73% ORR, 73% grade 3/4 AEs
Dabrafenib + Trametinib	BRAF-MEK inhibitors	Pembrolizumab	Melanoma	NCT02625337 [61]	Unknown	2	Pem and short-term/intermittent D/T—Median PFS 27.0 months vs. 10.6 months with Pem monotherapy
Dabrafenib + Trametinib	BRAF-MEK inhibitors	Spartalizumab	Melanoma	NCT04310397	Recruiting	2	2/2022
Dabrafenib + Trametinib	BRAF-MEK inhibitors	PDR001 (anti PD-1)	Melanoma	NCT02967692—COMBI-I [29]	Active, not recruiting	3	7/2023 ORR of 78%, including 44% complete responses (CRs)
Neoadjuvant Dabrafenib + Trametinib	BRAF-MEK inhibitors	Pembrolizumab	Melanoma	NCT02858921	Recruiting	2	11/2024
Trametinib +/− Dabrafenib	BRAF-MEK inhibitors	Nivolumab	Melanoma	NCT02910700—TRIDeNT [62]	Recruiting	2	12/2021 ORR 89%; PD1 refractory ORR 67%
Encorafenib + Binimetinib	BRAF-MEK inhibitors	Nivolumab	Melanoma with brain metastases	NCT04511013	Recruiting	2	6/2027
Encorafenib + Binimetinib	BRAF-MEK inhibitors	Nivolumab + Ipilimumab	Advanced melanoma after progression on targeted therapy	NCT03235245 [63]	Recruiting	2	2/2024 ORR 18.0% in the Nivo and 15.0% in the Ipi plus Nivo group
Encorafenib + Binimetinib	BRAF-MEK inhibitors	Nivolumab + Ipilimumab	Melanoma	NCT02631447 -SECOMBIT [64]	Active, not recruiting	2	12/2021 ORR was highest at 82.6% in Arm A (encorafenib + binimetinib until disease progression followed by Ipi + Nivo) with lowest toxicity
Encorafenib + Binimetinib	BRAF-MEK inhibitors	Pembrolizumab	Melanoma	NCT02902042	Active, not recruiting	1/2	2/2021
Encorafenib +/- Binimetinib	BRAF-MEK inhibitors	Nivolumab + Ipilimumab	Melanoma	NCT04655157	Not yet recruiting	1/2	7/2024
LXH254	BRAF/CRAF inhibitor	PDR001 (anti PD-1)	Advanced solid tumors	NCT02607813	Active, not recruiting	1	3/2021
Dabrafenib + Trametinib	BRAF-MEK inhibitors	Durvalumab (anti PD-L1)	Melanoma	NCT02027961 [65]	Completed	1	ORR 69.2%
Cobimetinib	MEK inhibitor	Atezolizumab	Melanoma, progressive disease on anti-PD-1	NCT03178851 [66]	Completed	1	ORR 36.4%, DCR 54.4%, DOR 12.7%, PFS 9.3% with cobimetinib prior to atezolizumab
Cobimetinib	MEK inhibitor	Atezolizumab + Bevacizumab	Melanoma with brain metastases	NCT03175432	Recruiting	2	6/2021
Binimetinib	MEK inhibitor	Nivolumab	Melanoma	NCT04375527	Recruiting	2	6/2023
TAK-580	Pan-RAF inhibitor	Nivolumab	Melanoma	NCT02723006	Terminated	1	Futility met
**Oncogenic Driver and Tyrosine Kinase Inhibitors**							
Imatinib	Multiple TKI	Ipilimumab	Melanoma, GIST	NCT01738139 [67]	Recruiting	1	Safe, without clear signal for synergy
Imatinib	Multiple TKI	Pembrolizumab	Melanoma w/ C-KIT mutation	NCT02812693	Withdrawn	1/2	Poor accrual
BMS-908662	RAF kinase inhibitor	Ipilimumab	Melanoma	NCT01245556	Completed	1	7/2012
Ibrutinib	BTK inhibitor	Pembrolizumab	Melanoma	NCT03021460	Recruiting	1	2/2021
ARRY-614	p38 MAPK and Tie2 inhibitor	Nivolumab + Ipilimumab	Melanoma	NCT04074967	Recruiting	1/2	11/2021
Capmatinib + Robociclib	MET inhibitor, CyclinD1/CDK4/6 inhibitor	PDR001 (anti PD-1)	Melanoma	NCT03484923	Recruiting	2	6/2022
Abemaciclib, Merestinib	CDK4/6 inhibitor, MET inhibitor	LY3300054 (anti PD-1), LY3321367 (Anti TIM-3)	Advanced solid tumors	NCT02791334 [68]	Active, not recruiting	1	12/2021 Early results—dose limiting hepatotoxicity, one patient PR
Sonidegib	Hedgehog Pathway inhibitor	Pembrolizumab	Melanoma, Advanced solid tumors	NCT04007744	Recruiting	1	7/2022
Avadomide	Cereblon inhibitor	Nivolumab	Melanoma	NCT03834623	Recruiting	2	5/2023
Ceralasertib	Ataxia telangiectasia and rad3 inhibitor	Durvalumab	Melanoma	NCT03780608	Active, not recruiting	2	12/2022
APG-115	MDM2 inhibitor	Pembrolizumab	Melanoma, Advanced solid tumors	NCT03611868	Recruiting	1b/2	2/1/2022
**Anti-angiogenic Molecules**							
Axitinib	VEGFR 1-3, C-KIT, PDGFR	Nivolumab	Melanoma	NCT04493203	Suspended	2	
Axitinib	VEGFR 1-3, C-KIT, PDGFR	Toripalimab (anti-PD-1)	Melanoma	NCT03941795	Recruiting	2	12/2022
Axitinib	VEGFR 1-3, C-KIT, PDGFR	Toripalimab (anti-PD-1)	Melanoma	NCT03086174 [69]	Completed	1b	48.3% ORR, median PFS 7.5
Lenvatinib	Multiple TKI—VEGFR1-2, FGFR1-4, PDGFR, KIT, RET	Pembrolizumab, Quavonlimab (anti CTLA-4)	Melanoma	NCT04700072	Not yet recruiting	1/2	4/2020
Lenvatinib	Multiple TKI—VEGFR1-2, FGFR1-4, PDGFR, KIT, RET	Pembrolizumab, Quavonlimab (anti CTLA-4), Vibostolimab (anti-TIGIT)	Melanoma	NCT04305041	Recruiting	1/2	4/2030
Lenvatinib	Multiple TKI—VEGFR1-2, FGFR1-4, PDGFR, KIT, RET	Pembrolizumab, Quavonlimab (anti CTLA-4), Vibostolimab (anti-TIGIT)	Melanoma	NCT04305054	Recruiting	1/2	4/2030
Lenvatinib	Multiple TKI—VEGFR1-2, FGFR1-4, PDGFR, KIT, RET	Pembrolizumab	Melanoma	NCT03776136	Active, not recruiting	2	6/1/2021
Lenvatinib	Multiple TKI—VEGFR1-2, FGFR1-4, PDGFR, KIT, RET	Pembrolizumab	Melanoma	NCT03820986 [70]	Recruiting	3	8 responses, DCR 67.4%, Median PFS 4.2 months
Lenvatinib	Multiple TKI—VEGFR1-2, FGFR1-4, PDGFR, KIT, RET	Pembrolizumab	Melanoma	NCT04207086	Recruiting	2	3/2024
Cabozantinib	Multiple TKI—MET, VEGFR2, RET	Nivolumab	Advanced cancers and HIV	NCT04514484	Recruiting	1	11/2025
Apatinib	VEGFR-2 TKI	SHR1210 (anti PD-1)	Melanoma	NCT03986515	Recruiting	2	5/2022
Apatinib	VEGFR-2 TKI	SHR1210 (anti PD-1)	Acral Melanoma	NCT03955354	Recruiting	2	4/2021
Apatinib Temozolomide	VEGFR-2 TKI, DNA alkylating agent	SHR1210 (anti PD-1)	Acral Melanoma	NCT04397770	Not yet recruiting	2	2/2023
Anlotinib	VEGFR-2 TKI, other TKI	TQB2450 (anti PD-L1)	Acral Melanoma	NCT03991975	Recruiting	1b	12/2021
Trebananib	Angiopoietin-2 inhibitor	Pembrolizumab	Advanced solid tumors	NCT03239145	Recruiting	1b	8/2024 Results only in colorectal cohort
ENB-003	ETBR inhibitor	Pembrolizumab	Solid tumors	NCT04205227	Not yet recruiting	1/2a	11/2023
**Epigenetic Modifiers**							
Entinostat	HDAC inhibitor	Pembrolizumab	Melanoma	NCT03765229	Recruiting	2	6/2023
Entinostat	HDAC inhibitor	Pembrolizumab	Metastatic uveal melanoma	NCT02697630 [71]	Active, not recruiting	2	8/2023 (PR) observed in 3 patients resulting in an ORR of 10%; OS 11.5 months
Entinostat	HDAC inhibitor	Pembrolizumab	NSLC, expansion cohort in melanoma	NCT02437136	Unknown	1b/2	8/2019
Panobinostat	HDAC inhibitor	Ipilimumab	Melanoma	NCT02032810	Active, not recruiting	1	12/2021
Domatinostat	HDAC inhibitor	Pembrolizumab	Melanoma	NCT03278665	Recruiting	1b/2	12/2022
Domatinostat	HDAC inhibitor	Nivolumab + Ipilimumab	Melanoma	NCT04133948	Recruiting	1b	3/2024
Tinostamustine	HDAC inhibitor	Nivolumab	Melanoma	NCT03903458	Recruiting	1b	3/2024
Chidamide	HDAC inhibitor	Nivolumab	Melanoma	NCT04674683	Recruiting	3	10/2025
Abexinostat	pan-HDAC inhibitor	Pembrolizumab	Advanced solid tumors	NCT03590054	Recruiting	1b	4/2022
ACY-241	HDAC inhibitor	Nivolumab and Ipilimumab	Melanoma	NCT02935790	Completed	1	12/2017
BMS-986158	BET inhibitor	Nivolumab	Advanced solid tumors	NCT02419417	Recruiting	1	12/2023
Guadecitabine	DNA hypomethylating agent	Ipilimumab	Melanoma	NCT02608437 [72]	Unknown	1	ORR 26%, disease control rate 42%
Evofosfamide	Alkylating agent	Ipilimumab	Melanoma, other solid tumors	NCT03098160	Unknown	1	4/2019
Olaparib	PARP inhibitor	Pembrolizumab	Melanoma	NCT04633902	Not yet recruiting	2	12/2024
Azacitidine	Cytidine nucleoside analog	Pembrolizumab	Melanoma	NCT02816021	Recruiting	2	2/2026
**Immunomodulators**							
Epacadostat	IDO inhibitor	Pembrolizumab, INCAGN01876 (GITR inhibitor)	Advanced malignancies	NCT03277352	Completed	1/2	6/2020
Epacadostat	IDO inhibitor	Pembrolizumab	Melanoma	NCT02752074 [73]	Completed	3	Median PFS 4.7 vs. 4.9 in Pem only—no significance
INCB001158, Epacadostat	Arginase inhibitor, IDO inhibitor	Pembrolizumab	Advanced solid tumors	NCT03361228	Terminated	1/2	Based on emerging data with epacadostat and Pem
BMS-986205	IDO 1 inhibitor	Nivolumab +/− Ipilimumab	Melanoma	NCT04007588	Withdrawn, slow accrual	2	Slow accrual
BMS-986205	IDO 1 inhibitor	Nivolumab	Melanoma	NCT03329846	Active, not recruiting	3	8/2020
Indoximod	IDO 1 inhibitor	Nivolumab, Pembrolizumab, Ipilimumab	Melanoma	NCT02073123	Completed [74]	1/2	ORR 55.7% (39/70, 36 confirmed) with CR of 18.6% (13/70, all confirmed). Median PFS 12.4 months
Duvelisib	PI3K inhibitor	Nivolumab	Melanoma	NCT04688658	Not yet recruiting	1/2	6/2022
Eganelisib	PI3K inhibitor	Nivolumab	Advanced solid tumors	NCT02637531 [75]	Completed	1	Early results SITC 2020—22% ORR in patients who had been refractory to ICI
PLX3397	CSF-1 receptor inhibitor	Pembrolizumab	Melanoma, other solid tumors	NCT02452424	Terminated	1 2	Insufficient evidence of clinical efficacy
ARRY-382	CSF-1 receptor inhibitor	Pembrolizumab	Melanoma, other advanced solid tumors	NCT02880371	Completed	1b/2	10/2019
SX-682	CXCR1/2—MDSC recruitment	Pembrolizumab	Melanoma	NCT03161431	Recruiting	1/2	12/2021
RTA 408 (Omaveloxolone)	NRF2 activator	Nivolumab +/− Ipilimumab	Melanoma	NCT02259231	Completed [76]	1/2	ORR 27%, 6 partial response and 2 complete response
Aspirin	COX-2 inhibitor	Ipilimumab, Pembrolizumab	Melanoma	NCT03396952	Active, not recruiting	2	6/2024
L-NMMA	INOS inhibitor	Pembrolizumab	Melanoma, solid tumors	NCT03236935	Recruiting	1b	3/2021
Itacitinib	JAK1 inhibitor	Pembrolizumab	Advanced solid tumors	NCT02646748	Active, not recruiting	1b	12/2021
RGX-104	LXR/ApoE inhibitor	Nivolumab, Pembrolizumab, Ipilimumab	Advanced solid malignancies and lymphoma	NCT02922764	Recruiting	2	3/2021
MIW815	STING agonist	PDR001 (anti PD-1)	Advanced solid tumors	NCT03172936	Completed	1b	12/2020
**Other**							
CB-839	Glutaminase inhibitor	Nivolumab	Melanoma, RCC, NSCLC	NCT02771626	Completed	1/2	6/2020
Trigriluzole	Glutamate release inhibitor	Nivolumab, Pembrolizumab	Metastatic solid malignancies or lymphoma	NCT03229278	Completed	1	1/2020
Etrumadenant	Adenosine receptor antagonist	Zimberelimab	Advanced malignancies	NCT03629756	Active, not recruiting	1	9/2021

Trials with unpublished outcomes (either in press or at conference proceedings) have the expected date of completion listed. Abbreviations: BRAF (B-raf proto-oncogene); WT (wild type); PFS (progression free survival); MEK (mitogen activated protein kinase kinase); ORR (overall response rate); αPD-1 (anti-programmed cell death protein 1); AE (adverse event); CR (complete response); DCR (disease control rate); DOR (duration of response); OS (overall survival): TKI (tyrosine kinase inhibitor); GIST (gastrointestinal stromal tumor); BTK (Bruton’s tyrosine kinase); MAPK (mitogen-activated protein kinase); Tie2 kinase (tunica interna endothelial cell kinase); MET (c-MET tyrosine kinase) DLT (dose-limiting toxicity); PR (progression rate); MDM2 (murine double minute 2 homologue); VEGFR (vascular endothelial growth factor receptor); FDA (Food and Drug Administration); PDGFR (platelet-derived growth factor receptor); FGFR (fibroblast growth factor receptor); RET (RET proto-oncogene); DNA (deoxyribonucleic acid); ETBR (endothelial receptor B); HDAC (histone deacetylase); PARP (poly ADP-ribose polymerase); GITR (glucocorticoid-induced TNFR-related); IDO (indoleamine-2,3 dioxygenase); PI3K (phosphoinositide 3-kinase); ICI (immune checkpoint inhibitors); CSF (colony stimulating factor); CXCR1/2 (C-X-C chemokine motif receptor); MDSC (myeloid derived suppressor cell); NRF (nuclear factor erythroid); COX (cyclooxygenase); PR (partial response); INOS (inducible nitric oxide synthase); JAK (Janus kinase); LXR/ApoE (liver X nuclear receptor/apoplipoprotein E); RCC (renal cell carcinoma); NSCLC (non-small cell lung cancer).

## Data Availability

Data sharing not applicable.

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
