# Peer review of "Thinking Small: Small Molecules as Potential Synergistic Adjuncts to Checkpoint Inhibition in Melanoma"

_ijms, 2021, doi:10.3390/ijms22063228_

Round 1

Reviewer 1 Report

This review summarizes the trends of clinical trials on combination therapy of small molecule-based anti-cancer drugs and immune checkpoint inhibitors, and demonstrates the possibility of the combination therapy in melanoma.

  1. Table 1 may be revised since target molecules and inhibitors of the target molecules are both described in the Molecular Target column. "Molecular Targeting Mechanism" instead of "Molecular Target" and the addition of BRAF inhibitor for Vemurafenib may be suitable.
  2. Figure 2 may be revised to add histone acetylation in epigenetic modulation in accordance with the figure legend. 

Author Response

After the reviewer’s kind comments:

  • Table 1 has been revised (Page 6, line 187) with the title of the second column now “Molecular Targeting Mechanism”). The column widths of the table have been adjusted to accommodate the change in text length.  The target “BRAF” has been changed to “BRAF inhibitor” for the mechanism of multiple drugs in the table, in accordance with the changed column title.
  • In addition, in the “Immunomodulators” section, the term indoleamine-2,3 dioxygenase was abbreviated to IDO for 3 drugs, to improve table readability.

  • Figure 2 (Page 19, line 445) has been revised to show histone acetylation (Ac) (lower right-hand corner, section 5). The abbreviation “Ac” has been added to the figure legend, accordingly. 

Reviewer 2 Report

The manuscript entitled "Thinking Small: Small Molecules as Potential Synergistic Adjuncts to Checkpoint Inhibition in Melanoma" provides new insights on the role of small molecules and immune checkpoint in melanoma therapy. Overall, this manuscript is well written and should attract a broad range of readership but I still have some tips.

Comments

  • The author should elaborate more about the role of small antibiotics, prebiotics, or nutritive in melanoma therapies.
  • There should be a significant section of small molecule inhibitor's role in immune cell regulation like lymphocyte, NK cells, and MDSCs.
  • Small molecule inhibitors have known to modulate the tumor microenvironment and induce immunogenic cell death which enhances immunotherapy. Are small molecules similarly synergizes with ICI? The author should describe the role of SMs in the tumor microenvironment.

Author Response

After the reviewer’s kind and thoughtful suggestions, the following changes have been made:

  • In the section on “other potential synergistic therapies, including nutritive therapies” (page 18, line 424), we have elaborated on the importance of the gut and tumor microbiome in treatment responsiveness to immune checkpoint inhibition. 3 sentences and 3 citations were added to emphasize the potential synergistic role of antibiotic use, nutrition, and other similar therapies which may modulate the microbiome in concert with immune checkpoint inhibition.  The complexity and preclinical nature of much of this work was also noted, in line with the previously existing sentence that much of this work is preclinical in nature and remains to be investigated clinically. 
  • In the section on “Dual immunomodulation” (page 17, line 385), 3 sentences and 1 citation were added to emphasize the potential for regulating various immune populations with small molecules and how this may synergize with immune checkpoint inhibition in melanoma. The potential tumor micro-environment components which may be involved in this putative synergy were detailed.  A sentence (line 400) was also modified to emphasize that much of this work is preclinical, and the focus of this manuscript is on small molecular drugs which have already progressed to clinical testing.
  • The mechanistic synergy between small molecules and immune checkpoint inhibition in the tumor microenvironment is intended to be a focus of the text and highlighted in figure 2. We appreciate the reviewer’s comments and agree this is important to discuss in further detail and have made changes to the manuscript accordingly.  The importance of small molecular drugs in modulating the tumor microenvironment has been emphasized in the modification listed above in point 2.  In addition, two sentences were added to the discussion (page 19, line 472) to discuss the potential roles which small molecular drugs may play in augmenting immune checkpoint inhibition within the tumor microenvironment.  The phrase “which act in the TME and are” was added to line 492 of the discussion as well.